# External Match Load in Amateur Soccer: The Influence of Match Location and Championship Phase

**DOI:** 10.3390/healthcare10040594

**Published:** 2022-03-22

**Authors:** Mauro Miguel, Rafael Oliveira, João Paulo Brito, Nuno Loureiro, Javier García-Rubio, Sergio Jose Ibáñez

**Affiliations:** 1Training Optimization and Sports Performance Research Group (GOERD), Sport Science Faculty, University of Extremadura, 10003 Cáceres, Spain; jagaru@unex.es (J.G.-R.); sibanez@unex.es (S.J.I.); 2Sport Sciences School of Rio Maior, Polytechnic Institute of Santarém, 2040-413 Rio Maior, Portugal; rafaeloliveira@esdrm.ipsantarem.pt (R.O.); jbrito@esdrm.ipsantarem.pt (J.P.B.); nunoloureiro@esdrm.ipsantarem.pt (N.L.); 3Life Quality Research Centre (CIEQV), Polytechnic Institute of Santarém, 2001-902 Santarém, Portugal; 4Research Centre in Sport Sciences, Health Sciences and Human Development, 5001-801 Vila Real, Portugal

**Keywords:** external load, contextual variables, soccer, amateur, home match, away match

## Abstract

Assessment of the physical dimension implicit in the soccer match is crucial for the improvement and individualization of training load management. This study aims to: (a) describe the external match load at the amateur level, (b) analyze the differences between playing positions, (c) verify whether the home/away matches and if (d) the phase (first or second) of the championship influence the external load. Twenty amateur soccer players (21.5 ± 1.9 years) were monitored using the global positioning system. The external load was assessed in 23 matches, where 13 were part of the first phase of the competition (seven home and six away matches) and the other 10 matches belonged to the second (and final) phase of the championship (five home and five away matches). A total of 173 individual match observations were analyzed. The results showed significant differences between playing positions for all the external load measures (*p* < 0.001). There were higher values observed in the total distance covered for central defenders (*p* = 0.037; ES = 0.70) and in high-intensity decelerations for forwards (*p* = 0.022; ES = 1.77) in home matches than in away matches. There were higher values observed in the total distance (*p* = 0.026; ES = 0.76), relative distance (*p* = 0.016; ES = 0.85), and moderate-intensity accelerations (*p* = 0.008; ES = 0.93) for central defenders, in very high-speed running distance for forwards (*p* = 0.011; ES = 1.97), and in high-intensity accelerations (*p* = 0.036; ES = 0.89) and moderate-intensity decelerations (*p* = 0.006; ES = 1.11) for wide midfielders in the first phase than in the second phase of the championship. Match location and championship phase do not appear to be major contributing factors to influence the external load while the playing position should be used as the major reference for planning the external training load.

## 1. Introduction

The assessment and knowledge of the physical dimension implicit in the soccer match is crucial in the improvement and individualization of planning structures since specific protocols can be designed in accordance with these demands [1]. In this regard, Bourdon et al. [2] assert that the load monitoring process should assist the coaches’ decision-making regarding the players’ availability to train and compete in order to achieve the main objectives of performance and injury prevention [3,4,5]. As manifested by Zurutuza et al. [6], it is essential to individualize training as much as possible in order to strengthen collective training and thus optimize competition performance.

Over the years, several methods have been used to determine the physical profile of soccer players [7]. In this sense, the global positioning system (GPS) and inertial sensors in wearable devices are widely used to measure external loads [8,9,10,11,12,13,14], which are objective measures of work performed by an athlete during training or competition [2]. Based on the perspective that measuring loads relative to competition demands could be an advantageous strategy that coaches use within training periodization models [12], the influence of the playing position and contextual factors (e.g., home/away, opponent’s standard positions, match period) on the match load has been a subject of particular attention [12,13,15,16,17] and their effects are well-reported.

In elite soccer, match loads have been being assessed for several years [8,9,10,11,12,13,14]. However, at the amateur level, clubs and their coaches do not have sufficient resources to allow them to precisely monitor the training/match load. Based on the identified differences between amateurs and professionals, Dellal et al. [18] suggest an adequation of training for these athlete populations. At the amateur level where athletes associate sports practice with another professional activity, the use of instruments that provide accurate data, even if used in spaced-out periods of the sports season, will allow the collection of data that help coaches to carry out adequate management of loads.

As stated by García-Rubio et al. [19], identifying the interactive effects of contextual variables on performance indicators can enable better team preparation. In this respect, Springham et al. [20] exposed that, of the analyzed contextual variables, only the playing position and goal deficit were identified as predictors of match physical performance. Additionally, some studies [21,22] describe that match location can exert a confounding effect on match physical performance, where home matches place greater physical demands on players compared with playing away. The study by Oliveira et al. [23] confirms that match location can influence internal and external load data preceding home and away matches. Lastly, Reche-Soto et al. [13] consider that physical, technical, tactical, and psychological training should be planned in relation to match location and level of the opponent. Amateur soccer teams usually have a lower frequency of training sessions, which requires physical, technical, tactical, and psychological training to be planned with great thoughtfulness with regard to match location [24]. According to the contextual factors (home/away, first/second phase), it may be necessary to adjust the microcycle to induce optimal physiological and performance recovery before the match [25,26]. In this sense, it is essential to analyze the effect that match location and championship phase can have on effort intensity of players on amateur teams.

Thus, to better identify and understand the current physical match demands placed on players in different playing positions at the amateur level as well as the impacts of contextual variables, this study aims to (a) describe the external match load at the amateur level, (b) analyze the differences in the external match load between playing positions, (c) verify whether the home/away contextual factor impacts the external match load, and (d) determine whether there are differences in match load between the first and the second phases of the championship.

## 2. Materials and Methods

### 2.1. Experimental Approach to the Problem

This investigation follows an associative strategy [27] where an attributive variable is utilized and differences between groups are examined. It is a longitudinal observational study performed with amateur soccer players participating in an official Portuguese regional competition.

Match data were collected over the 2018/2019 competitive season from October to June. The standard competitive microcycle included a match (Sunday) and three training sessions (Tuesday, MD–5; Thursday, MD–3; and Friday, MD–2—according to Malone et al. [28], training sessions are classified in relation to the number of days before the next competitive match). Twenty-four matches were observed throughout the data collection period, one of which was excluded from the analysis because the match was interrupted due to weather conditions and completed days later. Out of the 23 analyzed matches, 13 were part of the first phase of the competition (seven home and six away matches) while the other 10 matches belonged to the second (and final) phase of the championship (five home and five away matches). In the first phase, eight teams competed for the qualification to the final phase (the top three to classify would get access to this phase) where they would play for promotion (the analyzed team classified in the first place). In the final phase, six teams competed for the top three places giving access to the higher division (the analyzed team classified in the second place).

Assessment of the demands of a soccer match at the amateur level will allow increasing the knowledge about this little studied context (in terms of training/match monitoring). Evaluating the match load according to match location and championship phase will allow assessing the influence of these variables on the game and the players, and through this type of analysis, planning structures can be properly adapted to the specificities and particularities of the competitive period.

### 2.2. Participants

Twenty amateur soccer players (age: 21.5 ± 1.9 years old; height: 174.5 ± 7.9 cm; body weight: 71.2 ± 7.6 kg; fat mass: 17.5 ± 3.9%) from the same team that participated in the Portuguese men’s soccer championship (regional level) were included in this analysis. Given the preliminary nature of our study, we applied a stringent inclusion criterion. Players were only included in the analysis if they participated in at least four full matches during the data collection period. All the players and coaches were informed about the research protocol, requisites, benefits, and risks, and their written consent was obtained before the start of the study. The study protocol was approved by the ethics committee of the local university (No. 67/2017) and performed according to the ethical standards of the Declaration of Helsinki [29].

The analyzed team played in the 1:4:3:3 formation throughout the season, with two defensive midfielders and one offensive midfielder (these three players are hereinafter referred to as central midfielders). All the analyses were conducted according to playing positions (goalkeeper, central defender, full-back, central midfielder, wide midfielder, and forward). Goalkeeper’s data were only used to describe the match load for this playing position and were excluded from the comparative analysis. A total of 173 individual match observations were analyzed: goalkeeper (GK; *n* = 3 players, *n* = 18 cases), central defender (CD; *n* = 4 players, *n* = 35 cases), full-back (FB; *n* = 4 players, *n* = 38 cases), central midfielder (CM; *n* = 5 players, *n* = 44 cases), wide midfielder (WM; *n* = 5 players, *n* = 28 cases), and forward (F; *n* = 3 players, *n* = 10 cases).

### 2.3. External Match Load

The data from the external match load were collected using portable 10 Hz GPS devices (PlayerTek, Catapult Innovations, Melbourne, Australia); each of them also incorporated a triaxial 100 Hz accelerometer. This type of GPS devices seems to be the most valid and reliable for use in team sports [30].

The PlayerTek inertial devices were turned on and placed in a specific customized vest pocket located on the posterior side of the upper torso fitted tightly to the body, as is typically used in matches. These devices were turned on 10 min before the start of the warm-up period. During the monitoring period, the GPS devices would always be placed and checked by the same coach of the team, and each player would always use the same device [31].

The running variables obtained from the GPS were the total distance covered (TDC, m), the relative distance covered (RDC, m/min), and the distance covered (m) at five different speed thresholds: walking/jogging distance (WJD), 0.0 to 3.0 m/s; running-speed distance (RSD), 3.0 to 4.0 m/s; high-speed running distance (HSRD), 4.0 to 5.5 m/s; very high-speed running distance (VHSRD), 5.5 to 7.0 m/s; and sprint distance (SpD), a speed greater than 7.0 m/s [32]. The total number of accelerations and decelerations in three zones was also analyzed: low-intensity (LI Acc./LI Dec.), 0.0 to 2.0 m/s^2^; moderate-intensity (MI Acc./MI Dec.), 2.0 to 4.0 m/s^2^; and high-intensity (HI Acc./HI Dec.), greater than 4.0 m/s^2^ [9]. Moreover, player load (PL) was also included as a global load indicator in volume (AU) and intensity (AU/min).

### 2.4. Statiscal Analyses

All the statistical analyses were conducted using SPSS for Windows statistical software package version 22.0 (SPSS Inc., Chicago, IL, USA). Initially, descriptive statistics were used to describe and characterize the sample. Shapiro–Wilk and Levene’s tests were conducted to determine normality and homoscedasticity, respectively. One-way ANOVA was used with Scheffe’s post-hoc method. One-way analyses of variance were used to compare all the dependent variables (external match load measures) across the playing positions. Student’s t-test was also used to compare data by match location (home/away contextual factor) and championship phase (first and second phases). The effect size with 95% confidence interval (ES 95% CI) statistic was calculated to determine the magnitude of effects. Furthermore, Hopkins’ thresholds for the effect size statistics were used as follows: ≤0.2, trivial; >0.2, small; >0.6, moderate; >1.2, large; >2.0, very large; and >4.0, nearly perfect [33]. Alpha was set at *p* ≤ 0.05.

## 3. Results

### 3.1. Description of the External Match Load by Playing Position

Description of the 15 dependent variables by playing position is presented in Table 1. The central midfielders were the players who showed the greatest total distance covered (11,020 ± 720 m) and relative distance covered (112.7 ± 7.5 m/min). The forwards were those who showed greater very high-speed running distance (667 ± 158 m) and sprint distance (299 ± 96.6 m). The forwards and the wide midfielders exhibited the largest number of high-intensity accelerations, 36 ± 7 and 35 ± 8, respectively. The forwards were also the ones with the largest number of high-intensity decelerations (49 ± 7). The central midfielders were those who presented with greater player load, both absolute and relative (476 ± 31.2 AU and 4.9 ± 0.3 AU/min, respectively).

### 3.2. External Match Load—Comparison between Playing Positions

Significant differences were found between playing positions for all the external match load measures (*p* ≤ 0.001) and can be observed in detail in Table 2. Regarding the total distance covered, the central defenders (CD) presented with a significantly smaller distance compared to the other playing positions (*p* = 0.006; ES = −1.07 to −2.40), except for the wide midfielders (W), *p* = 0.075. The central midfielders showed a larger total distance than the other playing positions (*p* = 0.000; ES = 1.20 to 2.40), except for the forwards (*p* = 0.996). Regarding the relative distance covered, it appears that the central midfielders had a greater relative distance covered than the other playing positions (*p* = 0.000; ES =1.20 to 2.44), with the exception of the forwards (*p* = 0.980); in relation to the distance covered at different speed zones, the central defenders and the full-backs showed significant differences in all the zones (*p* = 0.025; ES = −2.51 to 1.59), with the exception of the running-speed distance (*p* = 0.774). A tendency towards the higher-intensity speed zone was noted for the full-backs, while the walking/jogging distance had higher values among the central defenders.

The central defenders and the forwards presented with significant differences in all the speed zones (*p* = 0.025; ES = −4.07 to 1.83). The forwards presented with higher values for running categories than the central defenders. Despite the forwards presenting with the highest average values in very high-speed running distance and sprint distance, there were no significant differences compared to the full-backs (*p* = 0.400) who presented with slightly lower values. In the three “acceleration zones”, the central midfielders showed higher values than both the central defenders (*p* ≤ 0.001; ES = 0.98 to 2.00) and the wide midfielders (*p* = 0.030; ES = −0.69 to 1.82). The central defenders presented with significant differences with all the other playing positions in moderate- (*p* = 0.013; ES = −0.98 to −2.10) and high-intensity accelerations (*p* = 0.000; ES = −1.65 to −3.16) and in moderate- (*p* ≤ 0.011; ES = −1.09 to −3.14) and high-intensity decelerations (*p* = 0.000; ES = −2.20 to −3.93).

Concerning the player load (AU), the central defenders presented with significantly lower values than all the other playing positions (*p* = 0.000; ES = −1.34 to −3.44), except for the wide midfielders (*p* = 0.213). The central midfielders presented with significantly higher values than all the other playing positions (*p* = 0.012; ES = 1.35 to 3.44). The wide midfielders only had significant differences with the central midfielders (*p* = 0.000; ES = −0.97). The same results were verified for the relative player load (AU/min).

### 3.3. External Match Load—Home vs. Away and First vs. Second Championship Phase

Variations in the external match load by playing position, between home and away matches, and between the first and the second championship phases are presented in Table 3 and Table 4, respectively. In the match location, significant differences were observed in the total distance covered for the central defenders (*p* = 0.037; ES = 0.70) and in high-intensity decelerations for the forwards (*p* = 0.022; ES = 1.77)—higher values in home matches. In the championship phase, significant differences were observed in the total distance covered (*p* = 0.026; ES = 0.76), relative distance covered (*p* = 0.016; ES = 0.85), and moderate-intensity accelerations (*p* = 0.008; ES = 0.93) for the central defenders, in very high-speed running distance for the forwards (*p* = 0.011; ES = 1.97), and in high-intensity accelerations (*p* = 0.036; ES = 0.89) and moderate-intensity decelerations (*p* = 0.006; ES = 1.11) for the wide midfielders—higher values in the first phase of the championship.

## 4. Discussion

The four aims of the present study were as follows: (a) to describe the external match load at the amateur level, (b) to analyze the differences in the external match load between playing positions, (c) to verify whether the home/away contextual factor influences the external match load, and (d) to determine whether there are differences in match load between the first and the second phases of the championship. While the first aim was merely descriptive (shown in Table 1), another analysis allowed noting relevant findings. The second purpose of the study showed significant differences between playing positions in several load measures. Finally, the third and the fourth purposes of the study found a tendency towards higher values in home matches than in away matches and in the first phase of the championship in comparison with the second phase, respectively, although mostly insignificant.

In the external load measures that are possible to compare, the results presented some peculiarities in comparison with other recent studies in professional/elite teams [9,14,21,25,34,35,36,37,38,39,40,41]. Based on the comparison of our results with those studies, greater differences were observed in the distance covered in higher-speed zones (developed in the following paragraphs). If the TDC seems identical, this could demonstrate that at higher competitive levels, one of the differences is in the physical requirement involved in a match. Therefore, examining the high-intensity activity provides a valid insight into physical performance and its strong relationship with the training status [42,43].

### 4.1. External Match Load—Description and Comparisons between Playing Positions

In relation to playing positions, the results showed that the central midfielders were the players who would cover the longest distance during a match, which is in line with recent literature findings [9,31,36,41,43,44,45], followed by the forwards, the full-backs, and the wide midfielders. As in other studies [34,38,41], the forwards were the ones who had the longest SpD, followed by the full-backs and the wide midfielders. Other features have been presented in other studies [36,46]. For example, Ingebrigtsen et al. [47] registered that players in lateral playing positions sprint the longest distances and more often compared to centrally playing players, while Paraskevas et al. [48] report that full-backs cover more SpD and VHSRD compared to all the other positions.

Regarding accelerations and decelerations, our data indicate that the central midfielders were those that presented with the greatest quantity, mainly of low and moderate intensity. The forwards were the ones that presented with higher values in HI Acc. and HI Dec. Similar results were found by Modric et al. [44] who reported that central midfielders were the players who performed more accelerations and decelerations (in a greater quantity in tactical formations with three defensive players in comparison with tactical formations with four such players), but the most intense accelerations and decelerations were carried out by forwards and full-backs. In contrast, Ingebrigtsen et al. [47] found that players in lateral positions accelerated more than those in central positions.

In recent studies [13,45], central midfielders were the players who showed the highest PL, followed by forwards, full-backs, wide midfielders, and central defenders. It has also been observed that the position of a central defender is the one that presents an external match load profile with more significant differences compared to the other playing positions. Modric [36] asserted that this is understandable knowing that their technical roles (e.g., aerial duels, tackles, positioning, and interception of the balls passed to the attackers) are generally more focused on reactions or accelerations and high-speed running. On the other hand, full-backs and wide midfielders are the playing positions that have more similarities (full-backs and forwards also have an identical profile). Wide midfielders and forwards showed a similar profile in terms of accelerations, decelerations, and PL.

Based on the results obtained and comparing them with other studies [9,36,41,44,45,47], it is possible to determine that the differences between the competitive levels (amateur, semi-professional, and professional) are more visible in the most demanding external load measures (e.g., SpD), as well as in the existence of variability in the external load profiles presented by the different playing positions. Through this perspective, coaches and technical staff must continually evaluate their own methodologies, strategies, game styles, player characteristics, among other factors [43], because although there are references of the external load for the team (depending on the competitive level) and the players (depending on the position occupied), the search for the best possible performance implies the customization and individualization of the approaches to be developed with the athletes. Thus, it is essential that coaches know the requirements of their own way of playing [8] and, with this, plan and organize the external load to be applied during the microcycle, because the one-size-fits-all approach could provide tactically constrained physical data for selected positions that are challenging to interpret given the lack of contextualization [42]. At the same time, the identification of differences and similarities in the external load profiles between the different positions will allow the conception and selection of training exercises in which each group of players will participate. This statement is corroborated by Modric [36], who affirmed that training prescriptions in soccer should be based on the established requirements specific to the playing positions, thereby ensuring that players are more able to fulfill their game duties and tactical responsibilities throughout the competition. Therefore, all training exercises must be characterized knowing the physical demands of each exercise as well as the external load applied on each playing position [31,49]. Complementary analytical training exercises (no ball) can also be developed to ensure that everyone achieves the desired external load, which was also supported by Modric [48], who suggested that when using small-side games which do not require running at high speeds, coaches should include specific running drills that entail high-intensity running (e.g., high-speed running and sprinting) in the training sessions.

### 4.2. External Match Load—Home vs. Away Matches

In the present study, there was a clear trend towards higher values in the TDC, RDC, distance covered by speed zone, and PL for home matches; however, only the TDC for the central defenders was significant, with a moderate effect size. While Castellano [50], Lago [51], and Gonçalves [52] also found that the external load was higher when playing at home, Gonçalves [17] verified that the match load was not influenced by the match location. In the meantime, Teixeira [43] suggested that the quality of opposition and match outcome have a greater influence than match location. Paraskevas et al. [48] reported that a greater TDC was covered during home matches against weak opponents compared to home matches against strong opponents and the opposite during away matches.

The same tendency is observed with regard to accelerations, decelerations (only HI Dec. was significant for the forwards). Otherwise, although not significantly, the central midfielders had smaller numbers of accelerations and decelerations at low and moderate intensity in home matches. Aquino [21] and Lago-Peñas [22] found that home matches place greater physical demands on players, owing to the combined effects of crowd, travel, familiarity, referee bias, territoriality, specific tactics, and psychological factors [22]. Contrary to that, Reche-Soto [13] described that the external load was higher when the team was playing away, but clarified that the effect of the match location was not clear. Chena [53] advised that this finding could be more related to the emotional variables than to the requirement of training sessions during that week. In other scope and in line with our results, Gonçalves [17] found that external loads were not influenced by match location and asserted that each competition may present its own idiosyncrasies, which is the reason why data generalization should be performed with caution.

The trend towards higher external load values in home matches can be explained by the offensive and defensive strategies adopted by both teams [43,48,52,54] as well as the size of the field. The analyzed team played at a stadium with natural grass and large dimensions (meets the requirements for international competitions) in home matches, which is different from the experience of most amateur clubs in Portugal (artificial grass and medium dimensions, some of which do not meet the requirements for national competitions). According to Almeida et al. [54], the defensive strategies used by better teams imply more intense and organized collective processes in order to recover the ball directly from the opposing team. Gonçalves et al. [52] reported that the counterattacking/transitional styles may result in greater high-intensity activities while the possession style may increase the distance covered in lower-speed zones. While, on the one hand, home matches increase the playing area available for opposing counterattacks (they demand a greater radius of action from central defenders), on the other hand, away matches reduce the area available for demarcations and moves by forwards.

Despite the well-proven knowledge that the size of the playing areas influences the load imposed on the players in small-side games [55,56,57], in amateur soccer, there is a diversity of fields (type of grass and dimensions) that can impact the physical demands of the game itself.

In our opinion, despite the match location presents an unclear effect on the external load, this contextual variable should be a subject of attention from coaches [17,43]. Existing changes, even if only in some specific metrics and for specific playing positions, may require reflection about and consideration of the need to adjust the planning, meeting the competition’s requests for each playing position (individual approach) [42,47]. Our results should not be used to generalize the dynamics of the external load in home/away matches—once again, we suggest that a continuous assessment be made of the team itself and that the approaches be centered on the analysis of the data themselves.

### 4.3. External Match Load—First vs. Second Championship Phase

The data suggest that there was a clear trend towards higher values in the TDC, RDC, accelerations, decelerations, and PL in matches of the first championship phase. However only the TDC and RDC for the central defenders were significant, with a moderate effect size, as well as MI Acc. for the central defenders with a moderate effect and for the wide midfielders in HI Acc. and MI Dec., also with a moderate effect. Curiously, in relation to the distance covered by speed zone, an inverse trend was observed. Lower values of VHSRD and SpD in the matches of the first phase of the championship (however, the forwards had a significantly higher VHSRD, with a large effect), which is contrary to Ingebrigtsen [47] results, who observed a shift towards more walking and fewer high-intensity locomotor activities during a match towards the end compared to the start of the season. Although not significant, the trend towards higher values of SpD and VHSRD in the second phase of the championship also contradicts the results by Springham [38] who claimed that the most notable decreases in performance were observed in sprint performance indices for which the greatest reductions were observed in full-backs, central midfielders, and wide midfielders. Additionally, Teixeira [43] suggested that contextual factors (quality of opposition and match outcome) and their changes seem to differ between playing positions.

In our study, the analysis of the championship phase required a careful reflection because of a simultaneous association between different opponents’ levels (in the second phase the teams had a similar level, with no weak teams) and periods of the competitive season. If, on the one hand, the confrontation with opponents of a higher qualitative level could cause a greater physical demand [48,52], on the other hand, the course of the competitive period itself leads to the loss of physical availability. According to Springham [38] results, all the physical performance indices decreased across the season; decreases in match physical performance indices in all the positions were observed. This author claimed that the cross-season decreases in match physical performance observed might be explained by longitudinal fatigue. Additionally, the absence of quality of opposition as a predictor variable for match physical performance is somewhat surprising as players are reported to complete more high-intensity activity and high-speed running when playing against high-quality opposition as opposed to low-quality opposition [20]. Conversely, Aquino [21] found that matches against weak opposition place greater physical demands on players.

In our opinion, the general and significant decreases observed may be associated simultaneously with the period of the season, as well as the type of opposition faced in the first and the second phases of the championship. Firstly, at the level (amateur) where the existing resources to help with physical recovery are scarce, the accumulation of trainings and matches throughout the competitive season can cause a decrease in physical freshness of the players. Finally, the offensive and defensive constraints posed by the opponents can assume co-responsibility for these results (mainly on the trend towards increasing SpD values) because a confrontation with opponents of greater value implies more moments of defensive and offensive transitions, moments of the game that demand fast and intense displacements on the part of all the players [52]. More specifically, while in the first phase the team dominated the games, continuously installing itself in the offensive midfield, in the second phase, this domain was not so evident, and the game assumed a more balanced nature.

### 4.4. Limitations

The main limitation of this study is the fact that only one team was observed, which is a very common obstacle in studies with soccer players [8,58]. In addition, the small sample size and the existence of other contextual factors such as the match result or the quality of opponents were not considered in the analysis and could provide better insights. Secondly, we only included the players who participated in full matches, which excludes the contribution of the other players who entered during in the game. All the matches played home were on natural grass, and of the 11 matches played away, 10 were on artificial grass and one—on natural grass. Finally, for us, it is difficult to associate the trend towards the increase in the external load in home matches only with the location when there are other variables that change with the match location.

## 5. Conclusions

These findings are novel and provide relevant information about the external match load in amateur soccer that can promote the reflection of coaches about the periodization and organization of training sessions at this competitive level:Match location and championship phase are not major contributing factors influencing the external load. Although there are several considerations regarding the influence of contextual variables on the match load, these should not be generalized and require their own evaluation and adoption of strategies based on them.The position occupied by players prevails as the most important factor in determining the load imposed by the competition, which is the reason why coaches must evaluate and characterize their game model in order to determine the physical demand imposed on the team and each player. Through the analysis of collective and individual match loads, it is possible to customize and individualize methodological approaches with regard to the management and regulation of the external training load.

## Figures and Tables

**Table 1 healthcare-10-00594-t001:** External match load by playing position (mean ± SD).

	Goalkeeper	Central Defender	Full-Back	Central Midfielder	Wide Midfielder	Forward	Team (a)
TDC (m)	4852 ± 592	9443 ± 547	10,129 ± 704	11,020 ± 720	10,003 ± 1004	10,906 ± 844	10,265 ± 963
RDC (m/min)	49.4 ± 5.7	96.5 ± 5.2	103.8 ± 7.2	112.7 ± 7.5	101.9 ± 9.8	110.9 ± 9.3	104.6 ± 9.8
WSJ (m)	4317 ± 545	6350 ± 375	5750 ± 373	6044 ± 438	5892 ± 218	3963 ± 2712	5914 ± 879
RSD (m)	324 ± 65.8	1717 ± 224	1954 ± 277	2463 ± 437	1855 ± 498	4484 ± 2705	2179 ± 957
HSRD (m)	178 ± 64.5	1048 ± 154	1551 ± 413	1832 ± 336	1501 ± 414	1494 ± 636	1505 ± 461
VHSRD (m)	31.9 ± 32.6	268 ± 74.7	636 ± 188	515 ± 166	579 ± 132	667 ± 158	503 ± 198
SpD (m)	0.9 ± 2.3	59.6 ± 41.8	239 ± 115	97.5 ± 77.3	190 ± 54.7	299 ± 96.6	148 ± 111
LI Acc. (n)	104 ± 23	210 ± 23	200 ± 23	244 ± 41	170 ± 39	206 ± 30	212 ± 40
MI Acc. (n)	73 ± 17	179 ± 23	231 ± 39	244 ± 38	212 ± 43	237 ± 39	221 ± 44
HI Acc. (n)	10 ± 6	16 ± 6	32 ± 7	29 ± 9	35 ± 8	36 ± 7	29 ± 10
LI Dec. (n)	94 ± 24	226 ± 40	214 ± 25	228 ± 33	192 ± 42	219 ± 39	219 ± 36
MI Dec. (n)	68 ± 17	147 ± 17	186 ± 38	218 ± 29	175 ± 33	205 ± 22	186 ± 40
HI Dec. (n)	10 ± 4	21 ± 7	39 ± 8	44 ± 12	41 ± 11	49 ± 7	37 ± 13
PL (AU)	207 ± 17.4	380 ± 22.3	420 ± 34.7	476 ± 31.2	399 ± 42.5	435 ± 22.5	425 ± 49.0
PL (AU/min)	2.1 ± 0.2	3.9 ± 0.2	4.3 ± 0.4	4.9 ± 0.3	4.1 ± 0.4	4.4 ± 0.3	4.3 ± 0.5

SD = standard deviation; TDC = total distance covered; RDC = relative distance covered; WJD = walking/jogging distance (0.0 to 3.0 m/s); RSD = running-speed distance (3.0 to 4.0 m/s); HSRD = high-speed running distance (4.0 to 5.5 m/s); VHSRD = very high-speed running distance (5.5 to 7.0 m/s); SpD = sprint distance (>7.0 m/s); LI Acc. = low-intensity accelerations (0.0 to 2.0 m/s^2^); MI Acc. = moderate-intensity accelerations (2.0 to 4.0 m/s^2^); HI Acc. = high-intensity accelerations (>4.0 m/s^2^); LI Dec. = low-intensity decelerations (0.0 to −2.0 m/s^2^); MI Dec. = moderate-intensity decelerations (−2.0 to −4.0 m/s^2^); HI Dec. = high-intensity decelerations (>−4.0 m/s^2^); PL = player load; AU = arbitrary units; m = meters; min = minutes; (a) all playing positions, excluding goalkeeper’s data.

**Table 2 healthcare-10-00594-t002:** External match load—comparison between playing positions, CI 95%, and ES.

	CD vs. FB	CD vs. CM	CD vs. WM	CD vs. F	FB vs. CM	FB vs. WM	FB vs. F	CM vs. WM	CM vs. F	WM vs. F
TDC (m)	−1234.6 to −138.3 **/moderate	−2107.0 to −1047.1 ***/very large	−1153.8 to 32.7	−2302.5 to −624.6 ***/very large	−1408.8 to −372.5 ***/large	−456.8 to 708.7	−1608.7 to 54.5	450.9 to 1582.2 ***/large	−706.2 to 933.2	−1765.0 to −41.0 */moderate
RDC (m/min)	−12.8 to −1.8 **/moderate	−21.5 to −10.8 ***/very large	−11.4 to 0.6	−22.9 to −5.9 ***/very large	−14.1 to −3.6 ***/large	−3.9 to 7.8	−15.5 to 1.3	5.1 to 16.5 ***/large	−6.5 to 10.0	−17.8 to −0.3 */moderate
WSJ (m)	48.5 to 1152.7 */large	−227.3 to 840.2	−139.3 to 1055.7	1542.0 to 3232.0 ***/large	−816.0 to 227.8	−729.3 to 444.6	948.9 to 2624.0 ***/large	−418.0 to 721.4	1254.9 to 2906.1 ***/large	1060.7 to 2797.0 ***/large
RSD (m)	−787.7 to 314.8	−1278.6 to −212.8 **/moderate	−734.4 to 458.7	−3610.4 to −1923.1 ***/very large	−1030.3 to 11.8	−487.4 to 684.6	−3366.5 to −1694.1 ***/very large	39.0 to 1176.7 */large	−2845.3 to −1196.8 ***/large	−3495.7 to −1762.1 ***/large
HSRD (m)	−771.7 to −234.9 ***/large	−1043.4 to −524.5 ***/large	−743.4 to −162.5 ***/large	−857.0 to −35.5 */large	−534.4 to −27.0 */moderate	−235.0 to 335.6	−350.1 to 464.2	54.0 to 607.9 */moderate	−63.6 to 739.1	−415.3 to 428.8
VHSRD (m)	−477.3 to −257.9 ***/very large	−352.6 to −140.5 ***/very large	−429.3 to −191.9 ***/very large	−566.4 to −230.7 ***/very large	17.4 to 224.8 */moderate	−59.6 to 173.7	−197.3 to 135.5	−177.3 to 49.1	−316.1 to 12.0	−260.5 to 84.5
SpD (m)	−240.1 to −119.3 ***/very large	−96.3 to 20.5	−195.8 to −65.0 ***/very large	−331.5 to −146.5 ***/nearly perfect	84.7 to 198.9 ***/large	−14.9 to 113.6	−151.0 to 32.4	−154.8 to −30.1 ***/large	−291.5 to −110.7 ***/moderate	−203.7 to −13.6 */large
LI Acc. (n)	−14 to 33	−56 to −6 */moderate	15 to 66 ***/large	−32 to 41	−66 to −21 ***/large	6 to 56 */moderate	−41 to 31	50 to 99 ***/large	3 to 74 */moderate	−73 to 1
MI Acc. (n)	−79 to −25 ***/large	−91 to −39 ***/very large	−62 to −5 */moderate	−99 to −17 **/very large	−38 to 12	−10 to 47	−46 to 35	4 to 59 */moderate	−33 to 47	−66 to 18
HI Acc. (n)	−21 to −10 ***/very large	−18 to −7 ***/large	−24 to −13 ***/very large	−28 to −12 ***/very large	−2 to 9	−8 to 3	−12 to 4	−12 to 0 */moderate	−16 to 1	−10 to 7
LI Dec. (n)	−14 to 37	−27 to 23	6 to 61 */moderate	−32 to 46	−38 to 10	−5 to 49	−44 to 34	9 to 62 **/moderate	−29 to 47	−67 to 14
MI Dec. (n)	−60 to −17 ***/large	−92 to −50 ***/very large	−52 to −4 */moderate	−91 to −24 ***/very large	−53 to −12 ***/moderate	−13 to 34	−52 to 14	21 to 66 ***/large	−19 to 46	−64 to 4
HI Dec. (n)	−25 to −11 ***/very large	−29 to −16 ***/very large	−28 to −12 ***/very large	−38 to −17 ***/very large	−11 to 2	−9 to 6	−20 to 1	−5 to 10	−16 to 5	−19 to 4
PL (AU)	−63.9 to −16.7 ***/large	−119.5 to −73.8 ***/very large	−45.4 to 5.7	−91.5 to −19.2 ***/very large	−78.7 to −34.0 ***/large	−4.7 to 45.5	−50.9 to 20.8	52.4 to 101.1 ***/very large	6.0 to 76.6 **/large	−72.6 to 1.7
PL (AU/min)	−0.7 to −0.2 ***/large	−1.2 to −0.8 ***/very large	−0.5 to 0.1	−0.9 to −0.2 ***/very large	−0.8 to −0.3 ***/large	−0.3 to 0.5	−0.5 to 0.3	0.5 to 1.1 ***/very large	0.1 to 0.8 **/large	−0.7 to 0.0

CD = central defender; FB = full-back; CM = central midfielder; WM = wide midfielder; F = forward; TDC = total distance covered; RDC = relative distance covered; WJD = walking/jogging distance (0.0 to 3.0 m/s); RSD = running-speed distance (3.0 to 4.0 m/s); HSRD = high-speed running distance (4.0 to 5.5 m/s); VHSRD = very high-speed running distance (5.5 to 7.0 m/s); SpD = sprint distance (>7.0 m/s); LI Acc. = low-intensity accelerations (0.0 to 2.0 m/s^2^); MI Acc. = moderate-intensity accelerations (2.0 to 4.0 m/s^2^); HI Acc. = high-intensity accelerations (>4.0 m/s^2^); LI Dec. = low-intensity decelerations (0.0 to −2.0 m/s^2^); MI Dec. = moderate-intensity decelerations (−2.0 to −4.0 m/s^2^); HI Dec. = high-intensity decelerations (>−4.0 m/s^2^); PL = player load; AU = arbitrary units; m = meters; min = minutes; CI 95% = 95% confidence interval; * *p* < 0.05; ** *p* < 0.005; *** *p* < 0.001. Effect size: ≤ 0.2, trivial; > 0.2, small; > 0.6, moderate; > 1.2, large; > 2.0, very large; and > 4.0, nearly perfect.

**Table 3 healthcare-10-00594-t003:** External match load—comparison between match location (home/away), mean ± SD, CI 95%, and ES.

	Central Defender		Full-Back		Central Midfielder		Wide Midfielder		Forward	
	Home	Away	Home	Away	Home	Away	Home	Away	Home	Away
TDC (m)	9646 ± 431	9271 ± 585	10,333 ± 583	9925 ± 769	11,163 ± 650	10,863 ± 775	10,214 ± 965	9793 ± 1032	10,948 ± 914	10,865 ± 875
	23.9 to 725.2/*/moderate		−40.9 to 857.1		−134.0 to 733.6		−355.1 to 1197.1		−1221.7 to 1387.3	
RDC (m/min)	98.0 ± 4.3	95.3 ± 5.6	105.3 ± 6.3	102.4 ± 8.0	113.2 ± 7.5	112.0 ± 7.6	103.1 ± 9.8	100.6 ± 10.0	110.2 ± 10.0	111.6 ± 9.7
	−0.9 to 6.1		−1.8 to 7.6		−3.4 to 5.8		−5.2 to 10.2		−15.8 to 13.0	
WSJ (m)	6465 ± 358	6253 ± 369	5771 ± 368	5729 ± 387	5973 ± 552	6120 ± 256	5921 ± 214	5863 ± 227	3852 ± 2768	4074 ± 2975
	−39.3 to 463.7		−206.5 to 290.5		−413.9 to 118.2		−113.9 to 228.7		−4412.6 to 3968.6	
RSD (m)	1760 ± 193	1681 ± 246	2033 ± 237	1874 ± 297	2482 ± 478	2442 ± 399	1956 ± 439	1755 ± 549	4542 ± 2925	4426 ± 2813
	−75.7 to 233.1		−17.9 to 335.3		−229.3 to 309.2		−184.9 to 587.4		−4068.3 to 4301.1	
HSRD (m)	1073 ± 165	1026 ± 146	1637 ± 375	1465 ± 440	1906 ± 289	1750 ± 370	1598 ± 313	1403 ± 487	1569 ± 725	1419 ± 609
	−59.2 to 154.6		−96.6 to 441.7		−44.9 to 357.5		−122.6 to 513.8		−826.1 to 1126.2	
VHSRD (m)	281 ± 86.4	257 ± 63.5	653 ± 201	619 ± 178	555 ± 179	470 ± 141	590 ± 142	567 ± 127	677 ± 174	657 ± 160
	−27.3 to 75.9		−90.4 to 159.2		−13.4 to 184.3		−81.3 to 127.5		−223.9 to 263.8	
SpD (m)	65.9 ± 49.6	54.3 ± 34.5	240 ± 110	239 ± 123	113 ± 98.8	80.2 ± 65.0	175 ± 39.7	205 ± 64.4	308 ± 132	289 ± 58.1
	−17.4 to 40.7		−76.3 to 77.4		−18.4 to 84.4		−71.8 to 11.3		−130.1 to 167.1	
LI Acc. (n)	212 ± 24	208 ± 23	202 ± 27	199 ± 19	240 ± 41	248 ± 41	175 ± 42	164 ± 38	211 ± 35	200 ± 27
	−13 to 19		−12 to 18		−33 to 17		−20 to 42		−34 to 57	
MI Acc. (n)	184 ± 19	175 ± 26	238 ± 41	223 ± 35	242 ± 36	246 ± 41	217 ± 38	207 ± 49	246 ± 35	227 ± 44
	−6 to 25		−10 to 40		−28 to 19		−24 to 44		−39 to 77	
HI Acc. (n)	14 ± 6	18 ± 5	33 ± 7	31 ± 7	31 ± 8	27 ± 9	33 ± 9	37 ± 8	35 ± 9	38 ± 4
	−7 to 0		−3 to 7		−2 to 9		−10 to 3		−13 to 7	
LI Dec. (n)	227 ± 48	225 ± 32	216 ± 24	213 ± 27	223 ± 38	234 ± 25	199 ± 40	185 ± 45	210 ± 41	228 ± 40
	−26 to 30		−14 to 19		−31 to 9		−19 to 47		−77 to 41	
MI Dec. (n)	149 ± 16	145 ± 18	194 ± 40	177 ± 34	216 ± 28	221 ± 31	179 ± 31	171 ± 36	215 ± 21	195 ± 21
	−9 to 15		−8 to 41		−22 to 14		−18 to 34		−11 to 49	
HI Dec. (n)	21 ± 7	21 ± 7	41 ± 8	38 ± 8	44 ± 11	43 ± 13	42 ± 11	41 ± 11	54 ± 4	44 ± 6
	−5 to 5		−2 to 8		−6 to 9		−7 to 10		2 to 18/*/large	
PL (AU)	384 ± 18.7	376 ± 24.8	427 ± 32.4	414 ± 36.5	479 ± 28.3	473 ± 34.5	409 ± 42.4	390 ± 42.0	436 ± 25.8	435 ± 21.8
	−7.1 to 23.7		−9.8 to 35.6		−13.0 to 25.3		−13.8 to 51.7		−33.7 to 36.0	
PL (AU/min)	3.9 ± 0.2	3.9 ± 0.2	4.3 ± 0.3	4.3 ± 0.4	4.9 ± 0.3	4.9 ± 0.3	4.1 ± 0.4	4.0 ± 0.4	4.4 ± 0.3	4.5 ± 0.2
	−0.1 to 0.2		−0.2 to 0.3		−0.2 to 0.2		−0.2 to 0.5		−0.5 to 0.3	

TDC = total distance covered; RDC = relative distance covered; WJD = walking/jogging distance (0.0 to 3.0 m/s); RSD = running-speed distance (3.0 to 4.0 m/s); HSRD = high-speed running distance (4.0 to 5.5 m/s); VHSRD = very high-speed running distance (5.5 to 7.0 m/s); SpD = sprint distance (>7.0 m/s); LI Acc. = low-intensity accelerations (0.0 to 2.0 m/s2); MI Acc. = moderate-intensity accelerations (2.0 to 4.0 m/s2); HI Acc. = high-intensity accelerations (>4.0 m/s2); LI Dec. = low-intensity decelerations (0.0 to −2.0 m/s2); MI Dec. = moderate-intensity decelerations (−2.0 to −4.0 m/s2); HI Dec. = high-intensity decelerations (>−4.0 m/s2); PL = player load; AU = arbitrary units; m = meters; min = minutes; CI 95% = 95% confidence interval; * *p* < 0.05. Effect size: ≤0.2, trivial; >0.2, small; >0.6, moderate; >1.2, large; >2.0, very large; and >4.0, nearly perfect.

**Table 4 healthcare-10-00594-t004:** External match load—comparison between the championship phases (first phase/second phase), mean ± SD, CI 95%, and ES.

	Central Defender		Full-Back		Central Midfielder		Wide Midfielder		Forward	
	First Phase	Second Phase	First Phase	Second Phase	First Phase	Second Phase	First Phase	Second Phase	First Phase	Second Phase
TDC (m)	9603 ± 550	9201 ± 461	10,052 ± 650	10,260 ± 797	11,067 ± 728	10,951 ± 724	10,090 ± 1102	9887 ± 889	11,266 ± 926	10,752 ± 832
	52.3 to 753.7/*/moderate		−690.1 to 273.3		−332.8 to 565.7		−596.0 to 1001.0		−848.6 to 1875.8	
RDC (m/min)	98.2 ± 5.0	94.0 ± 4.5	103.1 ± 6.9	105.1 ± 7.7	113.3 ± 7.7	111.8 ± 7.3	102.8 ± 10.9	100.6 ± 8.4	112.7 ± 9.9	110.2 ± 9.8
	0.8 to 7.5/*/moderate		−7.0 to 2.8		−3.1 to 6.2		−5.6 to 10.0		−13.0 to 18.2	
WSJ (m)	6439 ± 367	6216 ± 357	5777 ± 370	5703 ± 388	5958 ± 521	6167 ± 242	5833 ± 210	5970 ± 212	6008 ± 206	3087 ± 2833
	−32.3 to 477.8		−183.1 to 330.5		−475.9 to 56.9		−302.7 to 28.5		−987.3 to 6829.1	
RSD (m)	1772 ± 236	1636 ± 182	1911 ± 269	2028 ± 283	2463 ± 469	2462 ± 400	1988 ± 506	1679 ± 449	2251 ± 628	5441 ± 2700
	−16.3 to 288.0		−304.2 to 70.1		−273.6 to 273.9		−69.6 to 687.8		−6944.3 to 564.1	
HSRD (m)	1081 ± 158	998 ± 141	1507 ± 406	1626 ± 429	1901 ± 334	1732 ± 321	1551 ± 405	1433 ± 434	1918 ± 312	1312 ± 668
	−23.4 to 188.8		−401.7 to 163.5		−34.5 to 372.6		−209.2 to 446.3		−347.5 to 1558.7	
VHSRD (m)	256 ± 71.2	285 ± 79.0	622 ± 171	660 ± 219	531 ± 166	491 ± 168	559 ± 144	605 ± 116	837 ± 81.8	593 ± 120
	−81.2 to 23.2		−167.7 to 90.8		−63.1 to 143.3		−150.0 to 58.5		82.1 to 405.8/*/large	
SpD (m)	55.9 ± 43.8	65.2 ± 39.6	237 ± 94.5	244 ± 148	97.1 ± 79.1	98.0 ± 95.4	182 ± 47.0	201 ± 64.1	252 ± 31.9	319 ± 110
	−38.9 to 20.3		−87.1 to 72.1		−54.2 to 52.4		−62.4 to 23.8		−220.3 to 86.9	
LI Acc. (n)	215 ± 25	203 ± 18	197 ± 21	207 ± 25	245 ± 42	242 ± 41	180 ± 42	155 ± 31	204 ± 36	206 ± 30
	−4 to 28		−25 to 6		−22 to 28		−5 to 55		−53 to 38	
MI Acc. (n)	187 ± 21	167 ± 21	231 ± 32	230 ± 49	251 ± 38	233 ± 36	223 ± 46	199 ± 37	268 ± 36	223 ± 33
	6 to 35/*/moderate		−25 to 28		−5 to 41		−10 to 57		−10 to 99	
HI Acc. (n)	17 ± 7	16 ± 4	32 ± 7	33 ± 8	30 ± 9	28 ± 9	38 ± 8	31 ± 7	41 ± 8	35 ± 6
	−3 to 5		−6 to 4		−4 to 7		1 to 13/*/moderate		−4 to 16	
LI Dec. (n)	234 ± 44	214 ± 29	213 ± 22	216 ± 31	230 ± 39	225 ± 22	201 ± 47	181 ± 34	218 ± 58	219 ± 35
	−8 to 47		−20 to 15		−15 to 26		−13 to 52		−68 to 65	
MI Dec. (n)	150 ± 17	143 ± 18	182 ± 38	191 ± 39	224 ± 30	211 ± 28	189 ± 31	156 ± 26	222 ± 19	197 ± 20
	−6 to 19		−35 to 17		−5 to 30		10 to 55/*/moderate		−6 to 56	
HI Dec. (n)	21 ± 8	22 ± 6	39 ± 9	40 ± 7	44 ± 13	43 ± 10	42 ± 10	40 ± 12	47 ± 9	49 ± 7
	−6 to 4		−6 to 5		−6 to 9		−7 to 10		−14 to 10	
PL (AU)	385 ± 22.9	372 ± 19.6	416 ± 31.0	427 ± 40.5	478 ± 34.9	475 ± 25.8	405 ± 44.4	393 ± 40.9	441 ± 32.6	433 ± 19.5
	−2.2 to 28.3		−34.2 to 12.2		−16.8 to 22.3		−22.4 to 43.0		−28.4 to 46.2	
PL (AU/min)	3.9 ± 0.2	3.8 ± 0.2	4.3 ± 0.3	4.4 ± 0.4	4.9 ± 0.4	4.8 ± 0.3	4.1 ± 0.4	4.0 ± 0.4	4.4 ± 0.4	4.4 ± 0.2
	−0.1 to 0.3		−0.3 to 0.1		−0.2 to 0.3		−0.2 to 0.5		−0.5 to 0.4	

TDC = total distance covered; RDC = relative distance covered; WJD = walking/jogging distance (0.0 to 3.0 m/s); RSD = running-speed distance (3.0 to 4.0 m/s); HSRD = high-speed running distance (4.0 to 5.5 m/s); VHSRD = very high-speed running distance (5.5 to 7.0 m/s); SpD = sprint distance (>7.0 m/s); LI Acc. = low-intensity accelerations (0.0 to 2.0 m/s^2^); MI Acc. = moderate-intensity accelerations (2.0 to 4.0 m/s^2^); HI Acc. = high-intensity accelerations (>4.0 m/s^2^); LI Dec. = low-intensity decelerations (0.0 to −2.0 m/s^2^); MI Dec. = moderate-intensity decelerations (−2.0 to −4.0 m/s^2^); HI Dec. = high-intensity decelerations (>−4.0 m/s^2^); PL = player load; AU = arbitrary units; m = meters; min = minutes; CI 95% = 95% confidence interval; * *p* < 0.05. Effect size: ≤0.2, trivial; >0.2, small; >0.6, moderate; >1.2, large; >2.0, very large; and >4.0, nearly perfect.

## Data Availability

Data are available upon request to the contact author.

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
