# Peer review of "External Match Load in Amateur Soccer: The Influence of Match Location and Championship Phase"

_healthcare, 2022, doi:10.3390/healthcare10040594_

Round 1
Reviewer 1 Report
Overall, the text is well written, easy to understand, and the topic is of interest to team sports (Soccer). In an additional way, the knowledge of the physical demands resulting from matches in different conditions, can bring information about the teams' performance and help in the process of designing training programs.
However, the research topic is not exactly original, so I respectfully request the authors' attention to both specific parts and entire sections of the manuscript.
Please, find below the notes made by this reviewer:
Pg 2, line 85 - What does MD mean?
Pg 3, line 12 - Did the 1:4:3:3 team structure remain the same throughout the championship? Even though there were matches against different teams, player substitutions, and different contexts?
Pg 3, Line 121 - Would the "n" of 10 observations for the Forwards be enough to make the comparisons? Is the power of this sample compared to the number of teams and players in the championship representative?
Pg 3, Lines 122 to 132 - What is the frequency of acquisition of the equipment used?
What is the measurement error of the equipment?
How was the equipment calibrated for each field where the matches took place?
Pg 7 to 8 - The tables (2, 3 and 4) are in the discussion section!
Results and Discussion:
Even though it was conducted with amateur players, the higher external load in matches at home is not exactly the original finding. It would be interesting to discuss the results considering the current knowledge, presenting their significance beyond a comparison with available papers. This is also true when comparing different positions and league phases.
Conclusions:
I suggest rewriting the initial part of the conclusions in a more pointed way, showing what can be concluded from the experimental model, evolving to the reflections presented in this topic.
Author Response
Dear Reviewer,
First, thank you very much for your comments and suggestions that will certainly make the article more robust and capable, in terms of knowledge to be shared.
Best regards,
Mauro Miguel

Reviewer 2 Report
Thanks for the opportunity to review this manuscript, I believe that the authors of this manuscript focused on a significant topic. My comments are as followed.
Abstract
- Please add more details about the method into the Abstract.
- The second paragraph and third paragraph can be merged in the introduction, there is no need to separately present the second paragraph.
- Authors need to clarify the importance of precisely monitoring the training/match load at the amateur level in the introduction.
- The percentage of sex need to be added to the Methods.
- Please add more details about ethics reviewing, such as the code number.
- More information about the limitations of this study is needed.
Author Response

(The authors gave the same response as above.)

Reviewer 3 Report
Congratulations on the work done.
In detail, my comments are:
Introduction: in the objectives of the study, eliminate the word “influence”. You should write “effects” or “impacts”.
Statistical analysis
I have some doubts regarding confirmations of normality and homogeneity of data in such a small sample (including a sample smaller than 30 subjects and with several analysis variables). More adjusted statistical analyzes should have been applied for studies of the longitudinal type and indicate the reference analysis models for the data.
Results.
Care should be taken when phrases such as “and the same relationship is verified for the” are used in the results, and no test was indicated to study relationships between variables, nor was it indicated as an objective.
Discussion
Care must be taken with personal opinions in the discussion because all the information must be substantiated, and also with the possible associations attributed between variables when this was not the subject of analysis.
Limitations
Can be completed
Author Response

(The authors gave the same response as above.)

Round 2
Reviewer 1 Report
Nothing to add.
Author Response
Good afternoon dear Reviewer,
Once again, we appreciate your contribution.
Best regards,
Mauro Miguel
Reviewer 3 Report
Regarding studies with longitudinal data, it should be considered that:
There are variables that are repeated over time. Because there is a correlation between repeated observations, it will be necessary to use more complex statistical techniques. In longitudinal data we have two major dimensions to consider: individual and time.
With longitudinal data it is possible to calculate, for each period of time, statistics such as the mean value, standard deviation, but also the matrices of covariances and correlations between different periods of time.
I also want to point out that to use longitudinal data, one can resort to linear temporal trends or the analysis of response profiles (this approach, which is relatively simple but has two major limitations: The first is because the model becomes extremely complex as the time periods and explanatory variables increase. On the other hand, it does not take into account the fact that the time variable has a natural order, this is called natural tendency).
Author Response
Good afternoon dear Reviewer,
Once again, we appreciate your contribution. We are also grateful for the detailed explanation of the analysis models, it is valuable.
Best regards,
Mauro Miguel